# Paliperidone to Treat Psychotic Disorders

**Hormazd D. Minwalla** [1,*], **Peter Wrzesinski** [1], **Allison Desforges** [2], **Joshua Caskey** [2], **Brittany Wagner** [2], **Patrick Ingraffia** [2], **James C. Patterson II** [1], **Amber N. Edinoff** [1], **Adam M. Kaye** [3], **Alan D. Kaye** [4], **Omar Viswanath** [4,5,6,7] and **Ivan Urits** [4,8]

1   Department of Psychiatry and Behavioral Medicine, Louisiana State University Health Sciences Center, 1501 Kings Hwy, Shreveport, LA 71103, USA; pwrzes@lsuhsc.edu (P.W.); jpatte@lsuhsc.edu (J.C.P.II); aedino@lsuhsc.edu (A.N.E.)
2   School of Medicine, Louisiana State University Health Sciences Center, Shreveport, LA 71103, USA; adesfo@lsuhsc.edu (A.D.); jcaske@lsuhsc.edu (J.C.); bwagn1@lsuhsc.edu (B.W.); pingra@lsuhsc.edu (P.I.)
3   Department of Pharmacy Practice, Thomas J. Long School of Pharmacy and Health Sciences, University of the Pacific, Stockton, CA 95211, USA; akaye@pacific.edu
4   Department of Anesthesiology, Louisiana State University, Shreveport, LA 71103, USA; akaye@lsuhsc.edu (A.D.K.); viswanoy@gmail.com (O.V.); ivanurits@gmail.com (I.U.)
5   College of Medicine, Phoenix Campus, University of Arizona, Phoenix, AZ 84006, USA
6   Department of Anesthesiology, Creighton University School of Medicine, Omaha, NE 68124, USA
7   Valley Anesthesiology and Pain Consultants—Envision Physician Services, Phoenix, AZ 84006, USA
8   Southcoast Physicians Group Pain Medicine, Southcoast Health, Wareham, MA 02720, USA
*   Correspondence: hminwa@lsuhsc.edu; Tel.: +1-318-675-6619

**Abstract:** Purpose of Review: This is a comprehensive review of the literature regarding the use of paliperidone in the treatment of schizophrenia and schizoaffective disorder. It covers the background and presentation of schizophrenia and schizoaffective disorder, as well as the mechanism of action and drug information for paliperidone. It covers the existing evidence of the use of paliperidone for the treatment of schizophrenia and schizoaffective disorder. Recent Findings: Schizophrenia and schizoaffective disorder lead to significant cognitive impairment. It is thought that dopamine dysregulation is the culprit for the positive symptoms of schizophrenia and schizoaffective disorder. Similar to other second-generation antipsychotics, paliperidone has affinity for dopamine $D_2$ and serotonin 5-$HT_{2A}$ receptors. Paliperidone was granted approval in the United States in 2006 to be used in the treatment of schizophrenia and in 2009 for schizoaffective disorder. Summary: Schizophrenia and schizoaffective disorder have a large impact on cognitive impairment, positive symptoms and negative symptoms. Patients with either of these mental illnesses suffer from impairments in everyday life. Paliperidone has been shown to reduce symptoms of schizophrenia and schizoaffective disorder.

**Keywords:** paliperidone palmitate; antipsychotic agents; psychotic disorders; family characteristics; cognitive dysfunction; dopamine; serotonin

## 1. Introduction

Schizophrenia and schizoaffective disorder are two mental illnesses that have a deep impact on both their affected populations as well as our society as a whole. Multiple genetic and environmental factors contribute to the development of both disorders, leaving much of their etiology unknown [1]. The triad of schizophrenia includes cognitive impairment, positive symptoms and negative symptoms [2]. Drawing a fine line between this and schizoaffective disorder is rather difficult [3]. It is believed that schizoaffective disorder lies in the middle of the spectrum between schizophrenia and bipolar disorder, sharing characteristic symptoms of both [4]. Nonetheless, patients with either illness suffer great impairments to their everyday lives. Several studies have demonstrated that patients with schizophrenia or schizoaffective disorder experience similar social deficits in addition to a lower IQ when compared to control groups [4,5]. According to Charlson et al., "Despite

being a low prevalence disorder, schizophrenia ranked the 12th most disabling disorder among 310 diseases and injuries globally" [6]. It is estimated that unemployment rates run around 80–90% in schizophrenia patients and life expectancy is decreased 14.5 years on average [2,6]. Numerous factors contribute to the early mortality rate, with one of the main causes being the higher incidence of comorbid health conditions in schizophrenia patients [6]. A second major contributing factor is the increased lifetime suicide rate, which is estimated to be around 10% in this population [7]. The only known way to decrease this risk is through compliance with treatment [7].

Although there is much undiscovered about the exact pathophysiology of these disorders, what is evident is the severe economic burden they bring to our society [2]. The estimated expense for schizophrenia in the United States in 2013 was USD 155.7 billion [8]. This included the cost for direct healthcare, direct non-healthcare (mainly unemployment and caregiving) and indirect costs [8]. Schizophrenia patients between the ages of 25 and 54 bring on the bulk of the economic burden as this age range is when individuals are typically the most economically productive [6]. Their high unemployment rate leads to serious economic losses not only from the under-productivity of the patient but additionally from the under-productivity of the family [6]. Family members frequently have to care for the patient at home and pay for treatment. This combination results in considerable deficits in the health and welfare systems [6].

Together, these factors demonstrate a real need to find more effective treatments for these disorders. There are approximately 21 million people globally living with schizophrenia whose lives are hindered by the course of this illness [6]. Finding a treatment that patients will not only comply with but that has minimal side effects could drastically improve their individual lives, as well as relieve some of the societal burdens that accompany these illnesses. It has been suggested that current antipsychotic treatment might also contribute to the increased risk of suicide in patients, further strengthening our objective to find a better treatment course [6]. The aim of this article is to give a compiled overview of the drug paliperidone for the treatment of schizophrenia and schizoaffective disorder through exploring its pharmacological properties and clinical safety/efficacy.

## 2. Schizophrenia and Schizoaffective Disorder

### 2.1. Epidemiology

Schizophrenia and schizoaffective disorder share a very similar epidemiology, pathophysiology and presenting symptoms. However, there are some differences between them. The point prevalence of both disorders is low with schizophrenia being estimated at 0.28% and schizoaffective disorder at 0.11% [6,9]. Differences in prevalence among genders varies with studies. Some found that both schizophrenia and schizoaffective disorder had an equal distribution between sexes [6,10]. If a gender difference was found, it was trending towards schizophrenia being slightly more prevalent in males and schizoaffective disorder being slightly more prevalent in females [11,12].

Multiple studies have implicated the role of genetic abnormalities being a contributing factor for the development of schizophrenia ranging from polymorphisms in single genetic loci to copy number variants (CNVs) [1,2]. Hundreds to thousands of specific loci have been identified that are thought to contribute to the risk of developing schizophrenia [1,2]. Additionally, less frequent but high risk CNVs on genes encoding voltage-gated calcium channels, glutamate and dopamine receptors and components of post-synaptic density have been implicated as risk factors [2]. Despite the compelling evidence supporting these genetic differences in the etiopathogenesis of schizophrenia, no one single genetic variation is exclusively specific, making them non-diagnostic [1]. There is little research on the specific genetics of schizoaffective disorder. Nonetheless, based on similar heritability estimates and risk factors, it is very likely that schizoaffective disorder shares some of the genetic variants found in schizophrenia [10,12].

In addition to genetic changes, other studies have demonstrated environmental relationships. Multiple groups have found that individuals being born during the winter/early

spring months are at an increased risk for developing schizophrenia [13,14]. Individuals living in urbanized areas have also been demonstrated by multiple studies to have a considerably increased risk of developing schizophrenia and schizoaffective disorder [2,10,15]. Both of these increased risks have been proposed to possibly be the result of either in utero infection and/or other environmental exposures of the fetus [15]. Seasonal variation, as well as the high population in urban areas, could also be contributing factors to either susceptibility circumstance [15]. More recent evidence has been found to support the association between cannabis use and an increased risk of psychotic illness, including but not limited to schizophrenia and schizoaffective disorder [16,17]. A strong association with schizoaffective patients and a family history of affective disorders has also been affirmed [18].

As previously mentioned, although there are several risk factors leading to increased susceptibility of developing schizophrenia and schizoaffective disorders, no one factor is solely responsible [15]. These illnesses are highly polygenic and depend on a diverse interaction between the environment, psychology and social surroundings individual to each patient [15].

### 2.2. Pathophysiology

Dopamine dysregulation is one of the most widely accepted pathophysiological processes leading to the positive symptoms seen in schizophrenia [2]. The basis of this belief stems from the fact that schizophrenia patients seem to have a hypersensitivity to dopamine-like drugs, giving them enhanced symptoms such as delusions and hallucinations when compared to control groups [2,19]. This is additionally supported by evidence that current treatment with antipsychotics that block dopamine receptors sufficiently reduce hallucinations and delusions seen in these patients [1]. While it is clear that dopaminergic dysfunction plays a large role in the genesis of both schizophrenia and schizoaffective disorder, it is not the only factor responsible [20]. Abnormalities of glutamatergic function have also been implicated for contributing to the disturbances seen in these disorders [21]. Evidence from multiple studies investigating genetics, imaging, NMDA receptor (NMDAR) antagonists and treatments that increase NMDAR function in schizophrenia patients all support the notion of NMDAR hypofunction contributing to symptoms [22].

Multiple studies researching brain imaging identified that both schizophrenia and schizoaffective disorder patients had reduced grey matter volume [2,5,23]. Although these reductions were found to be more drastic in schizophrenia patients, schizoaffective patients still showed remarkable similarity with regards to having volume reductions in the same cortical areas as schizophrenia patients [5]. Both groups displayed grey matter reductions in the temporal lobe, medial frontal cortex, insula, hippocampus and cerebellum [5]. The grey matter reduction is progressive throughout the course of the illnesses, with notable activity affecting the left hemisphere and temporal lobe during the early stages [24].

### 2.3. Clinical Presentation

Schizophrenia and schizoaffective disorder share many of the same presenting symptoms, but they differ in their severity leading to a considerably better prognosis in schizoaffective disorder [3]. Both typically present in early adulthood and are characterized by complex psychopathology [2,6,11]. The main features of schizophrenia are positive symptoms, negative symptoms and cognitive impairment [2,25]. Positive symptoms consist of hallucinations, delusions and disorganized speech, while negative symptoms are characteristically reduced emotional expression, social withdrawal and impaired motivation [2,15]. The diminished cognitive functions experienced by patients include deficits in working memory and executive function, although there is "significant cognitive heterogeneity" among individuals [2,26].

In contrast, schizoaffective disorder is a much less stable diagnosis with much debate among clinicians and researchers [3]. According to DSM 5, schizoaffective disorder is diagnosed based on the following four criteria [27]. Criterion A states the patient must

experience symptoms of psychosis from criterion A of schizophrenia simultaneously with a major mood episode (manic or depressive) [27]. Criterion B requires two or more weeks of hallucinations or delusions in the absence of a major mood episode [27]. Criterion C states symptoms of a major mood episode must be "present for the majority of the total duration of the active and residual portions of the illness" [27]. Moreover, lastly, criterion D is the acknowledgment that none of the aforementioned disturbances are a result of another condition or substance use [27]. Much of the controversy with schizoaffective disorder is over the fact that it is described as an intermediate disorder between schizophrenia and bipolar disorder [3]. Several studies assessing cognitive functions and neuroimaging found schizoaffective disorder resembles schizophrenia significantly more than bipolar disorder [3,5,23]. IQ deficits and presenting symptoms (positive, negative and cognitive impairment) among schizophrenia and schizoaffective disorder were not different from each other but were found to be significantly different from the bipolar patients [3,5]. Additionally, brain imaging is consistent with schizoaffective disorder being more skewed towards schizophrenia than bipolar disorder [5].

### 3. Current Treatment of Schizophrenia and Schizoaffective Disorder

While dementia praecox, meaning premature dementia, was first described in 1887 by Emil Kraepelin [28], this was later redescribed by Eugen Bleuler as schizophrenia. Pharmacological treatment was not utilized until the 1950s, when chlorpromazine was synthesized [29]. The main pharmacologic treatment for schizophrenia and schizoaffective disorder is broken down into first-generation, second-generation and third-generation antipsychotics.

First-generation antipsychotics, also known as typical antipsychotics, work mainly by blocking the $D_2$ receptors in the brain, affecting the positive symptoms (e.g., hallucinations and thought disorders) of schizophrenia. These typical antipsychotics do not selectively block the $D_2$ dopamine receptors and can therefore have a wide range of side effects. Typical antipsychotics are further divided into high potency and low potency. The classification into high potency and low potency is based on "chlorpromazine equivalence," where haloperidol and fluphenazine are categorized as high potency and thioridazine is low potency, making it comparable with chlorpromazine. High potency antipsychotics are more likely to cause extrapyramidal symptoms (EPS) than low potency antipsychotics [29].

Second-generation antipsychotics, also known as atypical antipsychotics, are D2 and 5HT2A receptor antagonists. Studies have shown that 75–87% of patients presenting with first-episode psychosis respond to primary antipsychotic therapy within four weeks to one year [30–33]. Clozapine, the first FDA approved drug for treatment-resistant schizophrenia, was the catalyst to the discovery of the second-generation antipsychotics [34,35]. While it was the first drug of its class, it doesn't come without its many side effects. Many physicians will avoid using clozapine prior to the failure of other drug combinations due to the five black box warnings associated with the usage of the clozapine. Some risks included are agranulocytosis, orthostatic hypotension, myocarditis and seizures.

Third generation antipsychotics are the newest group and have been individualized based on their mechanism of action at dopamine receptors. Unlike the first- and second-generation antagonists, the third generations act as partial agonists at the $D_2$ receptors [29]. Cariprazine, approved in 2015, is a partial agonist at the $D_2$, $D_3$ and 5-$HT_{1A}$ receptors, with an affinity 10 times higher for the $D_3$ receptor than the $D_2$ receptor. This affinity has made this drug especially useful for the treatment of schizophrenia with dominant negative symptoms, a classically difficult to treat subset of patients [36].

Side effects of antipsychotics include EPS (e.g., dystonia, akathisia, parkinsonism), increased risk of neuroleptic malignant syndrome, weight gain, hyperprolactinemia and sedation [37]. These side effects are what guide a clinician to use one drug over another, as Huhn et al. found that efficacy differences between the antipsychotics are mostly gradual rather than discrete, while the side effect differences are more marked [37].

While treatment of schizophrenia and schizoaffective disorder remains anchored in pharmacologic therapy, non-pharmacological adjunctive treatments have proven to be ef-

fective. It has been shown that patients with less than six months duration of antipsychotic treatment who were enrolled in the model program NAVIGATE reported lower PANSS scores were more likely to remain in treatment, enjoy a better quality of life and participate more in work/school than usual care [38,39]. Patients with treatment-resistant schizophrenia can also benefit from non-pharmacologic modalities, such as cognitive behavioral therapy for psychosis, hallucination focused integrative therapy, repetitive transcranial magnetic stimulation and electroconvulsive therapy [40–45].

*Schizoaffective Disorder*

Schizoaffective disorder can be more difficult to diagnose, making it harder to study and, therefore, leaving it with far fewer FDA approved drug treatments. Currently, only paliperidone extended-release (ER) and long-acting injectable (LAI) forms, along with risperidone, have been shown to be effective in the treatment of schizoaffective disorder [46]. This is not surprising, considering paliperidone is an active metabolite of risperidone [46]. Most of the evidence provided for the treatment of this disorder is through indirect evidence, mainly from studies in patients with schizophrenia [47]. The only schizoaffective disorder specific studies with a large sample size were those conducted for paliperidone, which is how the drug got FDA approval for the treatment of this disorder [46].

## 4. Paliperidone Drug Info

Paliperidone is an atypical antipsychotic that is a major active metabolite of risperidone (5-hydroxyrisperidone), approved for use in the US in 2006 as daily oral extended-release tablets and 1-month or 3-month LAI formulations [48–50]. Paliperidone ER is indicated for treatment of schizophrenia in adults and adolescents ages 12–17 [48]. It is available in 1.5 mg, 3 mg, 6 mg, 9 mg and 12 mg [48]. The LAI form of paliperidone is indicated for the treatment of schizophrenia or schizoaffective disorder in adults as monotherapy or in conjunction with mood stabilizers, only after tolerance to oral paliperidone or risperidone has been demonstrated [49,50]. The 3-month injectable is to be used only after successful administration of the 1-month LAI [50]. The 1-month LAI is available in dosages of 39 mg, 78 mg, 117 mg, 156 mg, or 234 mg and the 3-month injectable is available in the higher dosages of 273 mg, 410 mg, 546 mg, or 819 mg [49,50]. Contraindications to the use of paliperidone include previous hypersensitivity reactions to paliperidone or risperidone [48–50]. Paliperidone, like all antipsychotics, is contraindicated for use in elderly patients with dementia-related psychosis, holding a boxed warning for increased mortality in this population [48–50]. Adverse events of paliperidone are consistent with other atypical antipsychotics due to dopamine blockade and effects at other neurotransmitter receptors. These include cerebrovascular disease (in the elderly), neuroleptic malignant syndrome, QT prolongation, extrapyramidal symptoms, tardive dyskinesia, weight gain, dyslipidemia, hyperglycemia, hyperprolactinemia, orthostatic hypotension, leukopenia, cognitive impairment and seizures [48–50]. No teratogenic effects have been demonstrated, but there is increased risk for EPS and/or withdrawal symptoms in neonates exposed during pregnancy [48–50]. Coadministration of paliperidone ER tablets or LAI with risperidone has not been studied [48–50].

## 5. Mechanism of Action

Paliperidone is in the benzisoxazole derivative class of atypical antipsychotics (including risperidone, iloperidone and paliperidone) and acts in accordance with others in the class [48]. While the full mechanism of atypical antipsychotics is yet to be fully realized, common features of the class include affinity for dopamine $D_2$ and serotonin 5-HT$_{2A}$ receptors [51]. Unlike typical antipsychotics, atypicals, in general, have a higher ratio of antagonism at 5-HT$_{2A}$ than $D_2$ receptors and have varying effects on other receptors [51]. In addition to 5-HT$_{2A}$ and $D_2$ antagonism, paliperidone has antagonistic effects at $\alpha_1$ and $\alpha_2$ adrenergic and $H_1$ histaminergic receptors [52]. It has no affinity for $M_1$ cholinergic or $\beta$ adrenergic receptors [52]. Positron emission tomography has shown paliperidone to oc-

cupy 70–80% of $D_2$ receptors in striatal and temporal cortex with a median effective dose of 2.38 mg/day and 2.84 mg/day, respectively [53]. Despite being the major active metabolite, paliperidone's affinity profile does not equal that of risperidone—paliperidone has a lower ratio of $5\text{-}HT_{2A}/D_2$ antagonism, lower affinity for $\alpha_1$ and $\alpha_2$ receptors and higher affinity for $H_1$ receptors [54]. Paliperidone receptor antagonism has been demonstrated in rats to affect serotonergic and noradrenergic signaling differently than risperidone in vivo [55]. Additionally, molecular signaling events following receptor binding have been shown to be distinct when comparing paliperidone and risperidone [56]. Paliperidone also has been demonstrated to induce mitochondrial protein expression changes in the prefrontal cortex similar to lithium, suggesting it may have mood stabilizing properties [57]. The molecular changes observed included increased proteins associated with receptor signaling, oxidative phosphorylation, neurotransmitter release and synaptic plasticity [57].

## 6. Pharmacokinetics/Pharmacodynamics

Paliperidone ER is designed with an osmotically controlled-release system which allows for continuous drug delivery after oral administration and no initial titration [48]. Paliperidone itself is insoluble in water with a volume of distribution of 487 L and it has an oral bioavailability of 28% [48,51]. Paliperidone ER reaches maximal concentration ($C_{max}$) in 24 h and increases in $C_{max}$ and the area under the drug concentration vs time curve (AUC) are observed after a high-fat or high-calorie meal [48,51]. Paliperidone ER demonstrates dose-response proportional AUC and $C_{max}$ in the recommended dose range [48,51]. The time to reach steady-state is 4–5 days and terminal half-life is 23 h [48,51].

The LAI paliperidone palmitate is deposited and slowly hydrolyzed to paliperidone over time. The 1-month and 3-month injectable forms reach $C_{max}$ in a median time of 13 days and 33 days, respectively, and both have higher $C_{max}$ when a deltoid injection is used rather than gluteal [49–51]. For this reason, the initial two doses are given in the deltoid muscle to rapidly achieve therapeutic concentration [49,50].

Paliperidone is minimally metabolized by the cytochrome P450 2D6 and cytochrome P450 3A4 enzymes; however, these are suggested to play a clinically irrelevant role overall [48,51]. Paliperidone is excreted 59% into urine unmetabolized and 11% into feces [48,51]. Divalproex sodium coadministration with paliperidone resulted in higher $C_{max}$ and AUC of oral paliperidone [48]. Carbamazepine coadministration decreased paliperidone by increasing renal clearance of paliperidone [48]. Renal impairment resulted in decreased clearance of paliperidone by 32% in mild impairment (CrCl from 50 mL/min to <80 mL/min), 64% in moderate impairment (CrCl from 30 mL/min to <50 mL/min) and 71% in severe impairment (CrCl from 10 mL/min to <30 mL/min) [48]. Mild to moderate hepatic impairment does not alter the plasma concentration of unbound paliperidone [48]. Effects of renal or hepatic impairment have not been directly studied for the LAI paliperidone palmitate.

## 7. Clinical Studies: Safety and Efficacy

### 7.1. Safety and Efficacy

Many studies have explored the efficacy and safety of paliperidone in patients with schizophrenia and/or schizoaffective disorder. Common measures of drug efficacy used in these studies include the Positive and Negative Syndrome Scale (PANSS), the Clinical Global Impression Severity (CGI-S) Scale, the Young Mania Rating Scale (YMRS), Hamilton Depression Rating Scale 21-item version (HDRS-21) and the Personal and Social Performance (PSP) Scale. Some studies also measured the time to relapse as a primary measure of clinical efficacy. In these studies, paliperidone was generally well tolerated; however, adverse effects were not uncommon and include headache, anxiety, insomnia, weight gain, suicidal ideation, nasopharyngitis, UTI and extrapyramidal symptoms.

Bossie et al. extrapolated data from a multiphase schizoaffective disorder study (NCT01193153) to examine the effect of paliperidone palmitate once-monthly injections (PP1M) in patients with recent-onset (≤5 years since first psychiatric diagnosis; $n = 206$) and chronic illness (>5 years; $n = 461$) versus placebo [58]. Multiple efficacy scales were

utilized in their study, including the PANSS, CGI-S, PSP, YMRS and HAM-D-21. It was found that both PP1M subpopulations displayed improvements in all scales used during the open-label PP1M acute and stabilization phases, with greater improvement noted in the recent onset subpopulation ($p \leq 0.022$). Relapse rates were higher in the placebo group compared to both PP1M subpopulations. In the recent onset subpopulation, the placebo group had a relapse rate of 30% and the PP1M participants had a relapse rate of 10.2% ($p = 0.014$; hazard ratio (HR): 2.8; 95% confidence interval (CI): 1.11–7.12; $p = 0.029$). In the chronic illness subpopulation, the placebo group had a relapse rate of 35.5% and the PP1M participants had relapse rate of 18.1% ($p = 0.014$; hazard ratio (HR): 2.8; 95% confidence interval (CI): 1.11–7.12; $p = 0.029$). Treatment-emergent adverse effects (TEAEs) occurred in over half of patients taking PP1M, with more noted in the chronic illness subpopulation (65.1%) compared to the recent onset (56.8%) subpopulation. The most common adverse effect was related to administration site conditions, such as pain. Other adverse effects reported were headache, insomnia, suicidal ideation, weight gain, akathisia, drug-induced parkinsonism, tremor and symptomatic prolactin related TEAEs, such as decreased libido and amenorrhea [58].

Fu et al. conducted a double-blind, randomized study evaluating the effects of paliperidone in 334 patients with schizoaffective disorder [59]. Paliperidone monotherapy and adjunctive paliperidone therapy were compared to placebo for 15 months. They found that PP1M significantly reduced the time to relapse for both treatment options ($p < 0.001$). The relapse risk was 3.38 times greater in the placebo group compared to PP1M monotherapy ($p = 0.002$) and 2.03 greater when compared to PP1M as adjunctive therapy ($p = 0.021$). The overall risk of relapse was 2.49 times greater for placebo (HR = 2.49; 95% CI: 1.55 to 3.99; $p < 0.001$), with an overall relapse rate of 33.5%. In contrast, the PP1M groups had relapse rates of 15.2%. Using the PSP scale, the researchers found that PP1M was superior to placebo in maintaining cognitive functioning. Common adverse effects found in this study were increased weight, insomnia, headache, nasopharyngitis and extrapyramidal symptoms [59].

Management of elderly patients with schizophrenia can be challenging, as they often have reduced liver and kidney function required to metabolize certain antipsychotic medications. Paliperidone, the primary metabolite of risperidone, is thought to be less likely influenced by changes in metabolism, leading to better outcomes in elderly patients with schizophrenia, while reducing treatment-emergency adverse effects. This understanding prompted Suzuki et al. to study the efficacy and safety of switching patients from risperidone to a pump-administered extended-release paliperidone formulation [60]. They enrolled 27 patients in their study, assigning 13 to the switch group and 14 to the control group (i.e., did not change from risperidone to paliperidone). The PANSS scores showed no significant difference between the two groups. This indicated, to the researchers, that paliperidone works equally well as risperidone in managing schizophrenia. Primary safety outcomes were evaluated using the Drug-induced Extrapyramidal Symptoms Scale (DIEPSS), the Drug Attitude Inventory Scale and prolactin levels. DIEPSS scores and prolactin levels had significantly greater reductions from baseline in the paliperidone group and the Drug Attitude Inventory Scale showed that elderly patients had more favorable views on paliperidone than risperidone. Additionally, patients in the paliperidone group required less biperiden when compared to the control group when EPS symptoms did arise even with similar risperidone-equivalent doses. The researchers thus concluded that paliperidone might result in superior safety and patient satisfaction in elderly patients [60].

Medication noncompliance is a major barrier for schizophrenia and schizoaffective disorder maintenance therapy. Like many patients with chronic medical conditions, patients with schizophrenia and schizoaffective disorder may not always comply with their antipsychotic medications because they have difficulty with daily oral therapy [61]. Therefore, longer-acting formulations can provide one means of optimized care for patients with chronic noncompliance problems. Hargarter et al. conducted a prospective, multicentral, open-label, 6-month study to see how patients with schizophrenia who failed

oral antipsychotic responded to the LAI formulation paliperidone palmitate [62]. Nearly 70% of the 212 patients enrolled in this study had clinical improvement in psychotic symptoms, as demonstrated by ≥30% improvement in mean PANSS scores ($p < 0.0001$) [61]. Another study, by Mauri et al., explored the effectiveness of switching to flexible doses of paliperidone ER from other antipsychotic regimens [63]. A total of 110 of the 133 patients were analyzed after the application of exclusion criteria such as inability to swallow oral medication. They found that patients had improvement in multiple scoring measures, including the PANSS, PSP and CGI-S scales when using paliperidone ER [63]. Furthermore, patients who have failed therapy on other long-acting or commonly used depot therapies can benefit from paliperidone palmitate injections. Schreiner et al. demonstrated that patients who switched from conventional depot antipsychotics ($n = 174$) or risperidone long-acting medications ($n = 57$) to paliperidone once monthly had significant reductions in mean PANSS scores, as well as improvement in symptom severity measured by the CGI-S [64]. The above studies also found that paliperidone formulations are generally well tolerated. Taken together, these studies support the notion that patients with schizophrenia and schizoaffective disorder can be better managed and suffer less from psychosis when taking paliperidone as a long-acting medication over other treatment modalities that indirectly promote noncompliance or had treatment failure on a different regimen.

Paliperidone injections can also be given once every three months. This formulation may be beneficial in preventing relapse of symptoms in schizophrenic and schizoaffective patients. Savitz et al. studied the efficacy and safety of paliperidone palmitate 3-month (PP3M) formulations for patients with schizophrenia in a randomized, multicenter, double-blind, noninferiority study [65]. PP3M was compared directly to the more conventional once-monthly paliperidone injections. Kaplan–Meier estimates showed that relapse rates were similar in those receiving PP3M and PP1M. Additionally, the pharmacokinetics between PP3M and PP1M were nearly the same, with no clinically relevant differences observed. The side effect profiles of the formulations were also similar, with weight gain being the most common treatment-emergent adverse effect [65].

Recent advancements in psychiatric research has discovered that levels of brain-derived neurotrophic factor (BDNF) could correlate with neuroprotection in schizophrenia, with higher levels indicating better outcomes [66]. A study by Wu et al. sought to quantify positive outcomes using antipsychotic treatment with risperidone or paliperidone with serum levels of BDNF and N400 latency and amplitudes. N400, an event-related brain potential component, recordings were performed to quantify cognitive functioning in schizophrenic patients. Both groups had increases in serum BDNF levels ($p < 0.01$) after 12 weeks of treatment, with no significant difference between the BDNF levels in the two treatment groups. N400 amplitudes also increased in both groups after treatment ($p < 0.01$). However, N400 latency periods were shorter with paliperidone therapy, compared to risperidone ($p < 0.01$), possibly indicating that paliperidone therapy leads to faster benefits in cognitive improvement. The researchers also used PANSS scores in their study, finding that it was significantly reduced in both groups ($p < 0.01$) after treatment [66]. One drawback of this study is that serum BDNF levels may be inferior to CNS BDNF measurements. Another drawback was the sample size of this study was small; larger samples will be needed in the future when using BDNF levels as a primary outcome. However, this study provides further support for paliperidone as an effective treatment for schizophrenia.

### 7.2. Comparative Studies

Researchers have also directly compared paliperidone to other treatment options in the management of schizophrenia and schizoaffective disorder. Alphs et al. compared once-monthly paliperidone palmitate to daily oral antipsychotic therapy in patients with schizophrenia using a randomized clinical trial [67]. Time to first treatment failure was the primary endpoint. This was defined by arrest/incarceration, hospitalization in a psychiatry ward, increased utilization of psychiatric services, suicide, or treatment discontinuation for reasons such as safety, tolerability, or efficacy. To measure time to treatment failure, the

Kaplan–Meyer method was utilized. Of the 450 patients enrolled in the study, 444 patients were included in an intent-to-treat analysis. They found that time to first treatment failure was significantly delayed in the paliperidone palmitate group. Over 15 months, the paliperidone group had treatment failure rates of 39.8%, whereas the oral antipsychotic group had treatment failure rates of 53.7%. The most common reasons were arrests and psychiatric hospitalizations for both groups. Common adverse effects in the paliperidone group were injection site pain, insomnia, weight gain, akathisia and anxiety [67].

Long-acting intramuscular versions of paliperidone have also been compared to long-acting haloperidol in the management of schizophrenia and schizoaffective disorder. McEvoy et al. found no significant difference between the efficacy of haloperidol and paliperidone [68]. Stroup et al. conducted a similar study that also demonstrated no significant difference between the two treatment options. However, Stroup et al. also found that haloperidol was associated with significantly longer time to efficacy failure in young patients (ages 18–45), whereas paliperidone was associated with significantly longer time to efficacy failure in older patients (ages 46–65). Stroup et al. noted that these age-related discrepancies warrant further investigation of these drugs in clinical practice [69]. Besides age, reasons to use one or the other may rely on side effect profiles. Prolactin levels and weight were demonstrated to increase more with paliperidone therapy compared to haloperidol. However, haloperidol was associated with a greater risk of extrapyramidal side effects, including akathisia [68,69]. Cost effectiveness should also be taken into consideration when choosing between haloperidol and paliperidone. A study by Rosenheck et al. concluded that paliperidone had slightly greater benefits in maintenance therapy of schizophrenia, but these benefits do not justify its high costs when haloperidol is markedly cheaper from the perspective of the healthcare system [39]. However, a first-generation antipsychotic is generally not preferred over second generations in the management of schizophrenia and schizoaffective disorder. Therefore, physicians must carefully consider the above findings when the decision to use a long-acting injectable is being made.

Aripiprazole is a third-generation antipsychotic that has demonstrated efficacy in treating schizophrenia and schizoaffective disorder. Its unique mechanism of action prompted Naber et al. to compare it to paliperidone. The researchers used the Heinrichs–Carpenter Quality-of-Life Scale (QLS) to compare once-monthly LAI formulations of aripiprazole to paliperidone in a 28-week randomized, noninferiority, rater-blinded, head-to-heady study. Aripiprazole was found to be superior to paliperidone in this regard [70]. Despite this, paliperidone is still a good option to consider when treating schizophrenia and schizoaffective disorder. This has been demonstrated numerous times, especially when compared to oral antipsychotics. Alphs et al. found that paliperidone palmitate injections reduce the risk of treatment failure when compared to oral antipsychotics, especially in recent-onset schizophrenia [71]. Another study by Alphs, Mao, Starr and Benson replicated these findings [72]. When compared to oral antipsychotics, paliperidone is more effective in delaying median time to treatment failure and reduces the number of treatment failures and psychiatric institutionalizations [72].

These studies, as well as numerous others, demonstrate that paliperidone is an effective therapeutic option in treating patients with schizophrenia and schizoaffective disorder. It is generally well tolerated, with weight gain being a common side effect. A question physicians often want to know is the type of formulation to give. The above works support that long-acting formulations are associated with better outcomes in schizophrenic and schizoaffective patients. When deciding between extended release tablets and long-acting injectables, physicians must take into consideration a patient's personal preference. However, research by Levitan et al. suggests that paliperidone once-monthly injections have shown greater benefits when compared to extended release tablets, especially early on in the disease process [73]. This drug should be considered in the management of schizophrenia and schizoaffective disorder. Table 1 is a summary of the clinical studies discussed in this section and Table 2 is a summary of the comparative studies.

**Table 1.** Clinical efficacy and safety.

| Author (Year) | Groups Studied and Intervention | Results and Findings | Conclusions |
|---|---|---|---|
| Bossie et al. (2017) [58] | Patients with chronic ($n = 461$) or recent onset ($n = 206$) schizoaffective disorder were treated with a 13-week open label acute treatment with PP1M, then 12-weeks stabilization with PP1M, then a 5-month double-blind relapse prevention, where patients were randomized to continue PP1M or withdrawal to placebo. | Both subpopulations showed significant improvement in mean psychotic, mood and function scores ($p \leq 0.022$). Relapse rates were higher with placebo than PP1M in the recent onset subpopulation (30% vs. 10.2%, $p = 0.014$) and the chronic illness subpopulation (35.5% vs. 18.1%, $p = 0.001$). The percentage of patients meeting all stabilization criteria was higher in the recent onset group (70.4%) than the chronic illness group (60%), $p = 0.010$. | Paliperidone is useful in managing schizoaffective disorder. It is especially beneficial in treating patients with recent onset disease and should be utilized clinically. |
| Fu et al. (2015) [59] | A total of 334 patients with schizoaffective disorder were randomized into a paliperidone once-monthly treatment group as monotherapy, or adjunctive treatment vs. placebo, first with a 13-week open-label phase, then a 12-week stabilization period, followed by a 15-month double-blind, relapse prevention phase. | Relapse risk was 2.49 times greater in placebo vs paliperidone once-monthly ($p < 0.001$). Paliperidone delays the time to relapse when added onto other medications regimens. The placebo group had a 3.38 times greater relapse risk than paliperidone monotherapy and a 2.03 times greater relapse risk than paliperidone adjunctive therapy ($p = 0.21$). | Paliperidone monthly significantly reduced episodic relapse in patients with schizoaffective disease vs. placebo. Paliperidone can be used as either a monotherapy or adjunctive therapy in patients with schizoaffective disorder. |
| Suzuki et al. (2013) [60] | A total of 27 inpatients with schizophrenia were switched to paliperidone therapy ($n = 13$) or maintained on risperidone ($n = 14$) and results were obtained at 12 weeks. | The PANSS score was the primary efficacy outcome measure; there was no significant difference between the paliperidone and risperidone groups. DIEPSS and prolactin levels were significantly decreased from baseline in the paliperidone group compared to the risperidone group ($-3.1$ vs. $-0.5$, respectively, $p = 0.0002$). Prolactin levels decreased more in the paliperidone group from baseline than the risperidone group ($p = 0.04$). Less biperiden was needed to manage EPS symptoms in the paliperidone group ($p = 0.006$). Patients reports more favorable views on paliperidone than risperidone using the Drug-Attitude Inventory Scale ($p = 0.0034$). | Paliperidone may result in superior safety outcomes and patient satisfaction in elderly patients with schizophrenia, when compared to risperidone. |
| Hargarter et al. (2014) [62] | A total of 212 patients with schizophrenia who failed oral antipsychotic therapy underwent a non-randomized, single-arm, multicentral, open-label, 6-month trial with once-monthly paliperidone injections. | Two-thirds of patients receiving paliperidone injections met the criteria for clinical response ($\geq$30% improvement in mean PANSS total score), $p < 0.0001$. | Schizophrenic patients respond to treatment with paliperidone. Patients who fail oral antipsychotic regimens could benefit by switching to paliperidone once-monthly injections. |

**Table 1.** *Cont.*

| Author (Year) | Groups Studied and Intervention | Results and Findings | Conclusions |
|---|---|---|---|
| Mauri et al. (2015) [63] | A total of 133 schizophrenic patients switched to paliperidone extended release PO and followed for 13-weeks. Patients were assessed at day 0, 14 days, 42 days and 91 days. | PANNS score decreased (from $88.98 \pm 10.09$ to $66.5 2 \pm 16.29$, $p < 0.001$). PSP and CGI-S scores also decreased ($p < 0.001$). Significant differences in these scores were found starting at week 2 and maintained throughout the trial. | Paliperidone ER shown to be efficacious. It can be considered in patients with schizophrenia. |
| Schreiner et al. (2015) [64] | A prospective, non-randomized, single-arm, multicentre, open-label, 6-month interventional study where schizophrenic patients switched from RLAT or oral antipsychotics ($n = 231$) to PP1M. | PANSS total score from baseline to last-observation-carried-forward were significantly reduced for both groups (from $-7.5$ to 10.6, $p \leq 0.01$ (BL to LOCF EP)). CGI-S scores also improved in the study participants ($p < 0.005$). Paliperidone is generally well tolerated. | Observed clinical benefits in schizophrenic patients taking paliperidone who failed other antipsychotic regimens. Paliperidone can be prescribed if other therapies have failed. |
| Savitz et al. (2016) [65] | Double blind, parallel group, multicenter, phase-3 study compared PP3M to PP1M in 1016 patients with schizophrenia in a 3-week screening period, 17-week open label phase; clinically stable patients were randomized to PP3m or PP1M for 48-week double-blind phase. | PP3M was non-inferior to PP1M with similar relapse rates in both groups (PP3M $n = 37$, 8%, PP1M $n = 45$, 9%); difference in relapse free rate (measured via Kaplan–Meier criteria) was 1.2%. No clinically relevant pharmacokinetic differences observed. Safety profiles similar, with weight gain being the most common side effect (double blind phase; 21% each). | PP3M showed similar efficacy as PP1M in preventing relapse in patients with schizophrenia. PP3M is a unique option and may be considered clinically for patients with schizophrenia. |
| Savitz et al. (2019) [74] | Eligible Latin American patients with schizophrenia were compared to rest-of-world patients (ROW); both groups received 17-week open-label PP1M stabilization, followed by two subsequent studies. Study A: patients randomized to PP1M or PP3M in a 48-week double blind treatment phase. Study B: patients entered a 12-week open label phase comparing PP3M to placebo. | Study A: relapse free percentage was similar in Latin America and ROW patients using PP1M or PP3M. Study B: median time-to-relapse was not significantly different between the placebo or PP3M Latin American patients. For the ROW group, median time-to-relapse for the placebo subgroup was 395 days and not estimable for the PP3M subgroup. | There is no significant difference in using PP1M or PP3M to prevent schizophrenia relapse in Latin American patients versus the rest-of-the-world. Latin American patients with schizophrenia can expect clinical benefits when taking paliperidone. |
| Wu et al. (2018) [66] | A total of 94 patients with first-time episodes of schizophrenia were randomly divided into risperidone or paliperidone treatment groups for 12 weeks. Serum BDNF levels, the latency and amplitude of N400 and PANNS scores were compared before and after treatment in the two groups. | Serum BDNF increased in both groups, no significant difference was found between the groups before and after treatment. After treatment, N400 amplitudes increased (from $4.73 \pm 2.86$ μv and $4.51 \pm 4.63$ μv to $5.34 \pm 4.18$ μv and 5.52 μv, $p < 0.01$). N400 latencies were shortened in the paliperidone group (from $424.13 \pm 110.42$ ms to $4.7.41 \pm 154.59$ ms, $p < 0.05$) under incongruent conditions. | Paliperidone and risperidone could increase serum BDNF levels in schizophrenic patients with first time episodes while also improving cognitive functions. This demonstrates that both drugs can improve the quality of life in schizophrenia patients and should be utilized clinically. |

**Table 2.** Comparative studies.

| Author (Year) | Groups Studied and Intervention | Results and Findings | Conclusions |
|---|---|---|---|
| Alphs et al. (2018) [71] | A 15 month randomized multicenter study of adults with schizophrenia and a history of incarceration; 450 patients assigned to PP1M ($n$ = 230) or PO antipsychotics daily ($n$ = 220). | PP1M associated with significant delay in time to first treatment failure when compared to PO antipsychotics (HR = 1.43; 95% CI = 1.09–1.88; log rank $p$ = 0.011). The failure rate for PP1m was 39.8%; 53.7% was the treatment failure rate in the oral antipsychotic group. Treatment failure was commonly due to arrest/incarceration (21.2% in PP1M; 29.4% in PO) and psychiatric hospitalizations (8.0% in PP1M; 11.9% in PO). The five most common TEAEs for PP1M were injection site pain, insomnia, weight gain, akathisia and anxiety. | Time to treatment failure was greater in the PP1M group compared to the PO antipsychotic daily group. |
| McEvoy et al. (2014) [68] | A total of 311 randomized schizophrenic or schizoaffective patients aged 18–65 placed in paliperidone palmitate ($n$ = 157) or haloperidol decanoate ($n$ = 154) followed for 24 months. | No statistically significant difference in rate of efficacy failure for paliperidone compared to haloperidol was found (HR = 0.98; 95%CI = 0.65–1.47). Paliperidone patients gained weight; haloperidol patients lost weight on average. Paliperidone increased prolactin levels, whereas haloperidol increased the risk of akathisia. | Paliperidone and haloperidol lead to similar outcomes in patients with schizophrenia and schizoaffective disorder. |
| Stroup et al. (2019) [69] | A total of 311 patients meeting the criteria for schizophrenia or schizoaffective disorder were randomly assigned to double-blinded treatment with haloperidol decanoate or paliperidone palmitate and followed for two years. | Age was correlated with different outcomes in different age groups. Young patients (ages 18–45) in the haloperidol group had longer time to relapse than participants in the paliperidone group. | Younger patients may respond better to haloperidol when compared to paliperidone. |
| Rosenheck et al. (2016) [39] | A total of 311 patients with schizophrenia or schizoaffective disorder were randomly assigned haloperidol or paliperidone once-monthly treatment for at least 24 months. | Paliperidone had 0.027 greater Quality Adjusted Life Years over 18 months ($p$ = 0.03), with greater quarterly medical costs including inpatient and outpatient treatment of USD 21,000/quarter. There is a 0.98 probability of greater cost-effectiveness when using haloperidol instead of paliperidone calculated using the Net Health Benefits analysis. | Paliperidone is not as cost-effective in the treatment of schizophrenia and schizoaffective disorder, even though it may be superior in managing symptoms. |
| Naber et al. (2015) [70] | A 28-week randomized trial comparing aripiprazole once-monthly and paliperidone once-monthly was conducted in 295 adult patients aged 18–60 years old with schizophrenia. | Statistically significant least squares mean difference in change from baseline to week 28 QLS total score demonstrated that aripiprazole is superior to paliperidone. Other ratings utilized gave further evidence to support this finding, including significant improvements in CGI-S and the Investigator's Assessment Questionnaire. | Aripiprazole once-monthly demonstrated superior efficacy and safety when compared to paliperidone. |
| Alphs et al. (2015) [67] | Schizophrenic patients with a history of criminal justice system involvement were randomly assigned paliperidone or oral antipsychotics and followed for 15 months. | Treatment failure hazard ratios for oral antipsychotics versus paliperidone were 1.73 (0.87–3.45; $p$ = 0.121) for recent onset patients and 1.37 (1.02–1.85, $p$ = 0.039) for chronic illness. | Paliperidone palmitate is associated with reduced risk of treatment failure compared to oral antipsychotic regimens. |

**Table 2.** *Cont.*

| Author (Year) | Groups Studied and Intervention | Results and Findings | Conclusions |
|---|---|---|---|
| Alphs, Mao, Starr and Benson (2016) [72] | Schizophrenic patients with a history of criminal justice system involvement were randomly assigned to monthly paliperidone injections (78–243 mg) or daily oral antipsychotic therapy in a 15-month prospective study. | Mean cumulative function of treatment failure events differed significantly in favor of paliperidone ($p = 0.007$) over oral antipsychotics ($p = 0.005$). | Paliperidone is superior to oral antipsychotics in delaying median time to treatment failure. |
| Levitan et al. (2016) [73] | Patients with schizophrenia were assigned to either extended release or injectable paliperidone once-monthly formulations and evaluated at 8 and 40 weeks. | At 8 weeks, PSP worsening, relapse, PANSS worsening and hospitalizations were significantly more associated with extended release paliperidone. At 40 weeks, relapse, PANSS, hospitalizations and CGI-S scales favored PP1M; however, these results were not significant. At both time intervals, safety outcomes were not significant amongst the groups. | PP1M was superior to extended release paliperidone tablets in the treatment of schizophrenia, especially earlier in the disease. |

## 8. Conclusions

The use of atypical antipsychotics remains central in the long-term treatment of schizophrenia and schizoaffective disorder. Relatively new as a long-acting injectable antipsychotic, paliperidone is of particular interest in promoting treatment adherence and reducing symptoms over longer periods of time. The 3-month paliperidone palmitate is currently the longest-acting antipsychotic available for use in schizophrenia and schizoaffective disorder. Its mechanism of action is similar to most other atypical antipsychotics, antagonizing $D_2$ and $5\text{-HT}_{1A}$ receptors. It has been shown to be effective and safe, compared to placebo, in the treatment of these disorders by various scoring measures in multiple studies. Paliperidone LAI is a favorable choice in patients who have failed oral antipsychotic regimens. Although some studies demonstrated slightly reduced efficacy compared to other antipsychotics, paliperidone has demonstrated reduced risk of treatment failure over time. Compared to its parent compound risperidone, paliperidone may be more favorable due to similar efficacy, increased safety and increased patient satisfaction. Paliperidone is not significantly affected by liver and renal function and is thus a good choice for consideration in patients with decreased hepatic or renal function. Further studies are required to evaluate combination long-term therapies and daily coadministration of other antipsychotics for schizophrenia or schizoaffective disorder.

**Author Contributions:** H.D.M., B.W., J.C., P.I., P.W. and A.D. were involved in writing of the manuscript. A.N.E., J.C.P.II, A.M.K., A.D.K., I.U. and O.V. were involved in manuscript editing. All authors have read and agreed to the published version of the manuscript.

**Funding:** This research received no external funding.

**Institutional Review Board Statement:** Ethical review and approval were waived for this study due to no human subjects being involved.

**Informed Consent Statement:** Not applicable.

**Data Availability Statement:** Data supporting the results above can be found on pubmed.

**Conflicts of Interest:** None of the authors have any conflict of interest to report in this project.

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
