# Peer review of "Paliperidone to Treat Psychotic Disorders"

_2035-8377, doi:10.3390/neurolint13030035_

Round 1

Reviewer 1 Report

The authors have done a nice job of summarizing the literature related to paliperidone. This literature spans nearly 20 years, so they have gone through great lengths to be as comprehensive as possible.

The article is written with care, and systematically goes through most of the relevant information from a clinical perspective. The authors present data in a balanced manner, and I do not see any evidence of bias. The data are presented in an objective manner.

There are no missing gaps that I can see from this review paper. I do have some rather minor comments which are provided below. I have no concerns recommending this for publication.

General: the authors use the term "adjuvant". This is a term that is used quite commonly in oncology, and its meant to convey therapy that is given after the initial treatment for cancers. It might be better for the authors to consider "adjunctive" instead, which is more appropriate for a psychiatry audience.

Page 2 line 49: approval for schizophrenia was 2006, but schizoaffective approval was not until 2009. Sentence implies that approval was given for both in 2006.

Page 8 line 174: Roman numerals are not preferred to describe DSM any longer. Suggest to change to DSM-5

Page 11 line 255: 12 mg dose form is also available.

Page 31 line 664: the citation for 48 contains some words which look very strange "These highlights do not include...". Please fix this citation.

Author Response

General: the authors use the term "adjuvant". This is a term that is used quite commonly in oncology, and its meant to convey therapy that is given after the initial treatment for cancers. It might be better for the authors to consider "adjunctive" instead, which is more appropriate for a psychiatry audience

Author notes: This has been changed to adjunctive

Page 2 line 49: approval for schizophrenia was 2006, but schizoaffective approval was not until 2009. Sentence implies that approval was given for both in 2006.

Author notes: these two years were noted for schizophrenia and schizoaffective disorder

Page 8 line 174: Roman numerals are not preferred to describe DSM any longer. Suggest to change to DSM-5

Author note:  changed to 5

Page 11 line 255: 12 mg dose form is also available.

author note: noted to have 12mg as well.

Page 31 line 664: the citation for 48 contains some words which look very strange "These highlights do not include...". Please fix this citation.

Author note: fixed to be more clear.

Reviewer 2 Report

The paper contains few minor formal negligence errors and omissions along with some more important issues related to the understanding of more basic concepts like what cognitive functions represent, the difference between disorganization and reality distorsion or essential pharmacodynamic knowledge of antipsychotics.

Overall, it appears that it is a mixture between a psychiatry lecture in the first part, and a literature review presenting the results of a number of clinical studies performed on paliperidone in its second part.

The complex and controversial topic regarding the use of antipsychotics in patients with dementia needs a more nuanced discussion.

167-168: "The main features of schizophrenia are positive symptoms, negative  symptoms, and cognitive impairment (2,25). Positive symptoms consist of hallucinations,  delusions, and disorganized speech, ..." .

There is no consensus that disorganized speech should be classified as a positive symptom. I suggest a more nuanced statement.

172:  change the singular into plural: cognitive functions is better than cognitive function as a complex set of cognitive processes are involved (attention control, problem solving, planning, monitoring, error correction, etc.)

174: the new reference is DSM-5 instead of DSM V (by contrast to DSM-IV-TR)

193: rephrase and correct Emile Kraepelin into Emil Kraepelin. He did not refer to schizophrenia but to dementia praecox which was later changed by  Eugen Bleuler into schizophrenia.

207-210: rephrase the whole paragraph. What is important to state is that second generation antipsychotics are D2 and 5HT2A receptor antagonists. The fact that they block other neurotransmitter receptors does not differentiate SGA from FGA.

259-261: The doses given in the article for the 1 and 3 month paliperidone LAI are for Invega Sustenna and Invega Trinza marketed in USA . In Europe, paliperidone LAI is marketed as Xeplion and Trevicta which come in different dosages such as 20, 50, 75, 100, 150 mg and 175, 263, 350, 525 mg respectively.

263: paragraph to be rewritten. The problem of prescribing of antipsychotics in patients with dementia is complex and still debated.  Evidence based medicine, expert opinion, practitioners and caregivers show divergent opinions.

322: I think the authors wanted to write clinical studies instead of clinical students

Author Response

167-168: "The main features of schizophrenia are positive symptoms, negative  symptoms, and cognitive impairment (2,25). Positive symptoms consist of hallucinations,  delusions, and disorganized speech, ..." .

Author note: this will be broken up to be clear but according to the DSM 5, disorganized speech is a positive symptom. Negative symptoms include avolition, apathy and flat facies. Delusions, hallucinations, disorganized speech/thought are positive symptoms.

193: rephrase and correct Emile Kraepelin into Emil Kraepelin. He did not refer to schizophrenia but to dementia praecox which was later changed by  Eugen Bleuler into schizophrenia

    author note: rephrased and corrected

207-210: rephrase the whole paragraph. What is important to state is that second generation antipsychotics are D2 and 5HT2A receptor antagonists. The fact that they block other neurotransmitter receptors does not differentiate SGA from FGA

    Author note: rephrased

259-261: The doses given in the article for the 1 and 3 month paliperidone LAI are for Invega Sustenna and Invega Trinza marketed in USA . In Europe, paliperidone LAI is marketed as Xeplion and Trevicta which come in different dosages such as 20, 50, 75, 100, 150 mg and 175, 263, 350, 525 mg respectively

   Author note: noted but not changed as this was focused on the medication in the USA.

174: the new reference is DSM-5 instead of DSM V (by contrast to DSM-IV-TR)

Author note: changed to DSM-5

Round 2

Reviewer 2 Report

93  Typing error, ad an x: praecox instead of preco